# Evaluation of Chemical Composition and Meat Quality of Breast Muscle in Broilers Reared under Light-Emitting Diode

**DOI:** 10.3390/ani11061505

**Published:** 2021-05-22

**Authors:** Francesca Bennato, Andrea Ianni, Camillo Martino, Lisa Grotta, Giuseppe Martino

**Affiliations:** 1Faculty of Bioscience and Technology for Food, Agriculture and Environment, University of Teramo, 64100 Teramo, Italy; fbennato@unite.it (F.B.); aianni@unite.it (A.I.); lgrotta@unite.it (L.G.); 2Istituto Zooprofilattico Sperimentale dell’Abruzzo e del Molise G. Caporale, Via Campo Boario, 64100 Teramo, Italy; c.martino@izs.it

**Keywords:** light-emitting diode, broiler, breast meat, fatty acids, volatile compound

## Abstract

**Simple Summary:**

The present study was designed to investigate the role of three different light-emitting diode (LED) light color temperatures (Neutral, Cool, and Warm) on the growth performance, carcass characteristics, and breast meat quality of broilers. No significant differences were observed in carcass yield in any of the experimental conditions. The changes observed in physical and chemical properties of breast meat samples suggest that LED light was not able to modify the quality of the products; therefore, it could represent a good alternative technology to traditional light sources.

**Abstract:**

The present study was designed to investigate the role of three different light-emitting diode (LED) light color temperatures on the growth performance, carcass characteristics, and breast meat quality of broilers. In our experimental condition, 180 chicks were randomly distributed into four environmentally controlled rooms (three replicates/treatment). The experimental design consisted of four light sources: neon (Control), Neutral (Neutral LED; K = 3500–3700), Cool (Cool LED; K = 5500–6000), and Warm (Warm LED; K = 3000–2500). Upon reaching the commercial weight (3.30 ± 0.20 kg live weight), 30 birds from each group were randomly selected, and live and carcass weight were evaluated to determinate the carcass yield. Following the slaughtering, samples of hemibreast meat were collected from each group and analyzed for physical and chemical properties, fatty acids composition, and volatile compounds. Live weight and carcass weight were negatively influenced by the Warm LED; however, no significant differences were observed in carcass yield in any of the experimental conditions. Higher drip loss values were detected in breast meat samples obtained by broilers reared under Neutral and Cool LEDs. In regard to the meat fatty acids profiles, higher polyunsaturated fatty acids (PUFA) values were detected with the Warm LED; however, the ratio of PUFA/saturated fatty acids (SFA) did not change in any group. The evaluation of volatile profiles in cooked chicken meat led to the identification of 18 compounds belonging to the family of aldehydes, alcohols, ketones, and phenolic compounds, both at 0 (T0) and 7 (T7) d after the cooking. The results of the present study suggest that the LED represents an alternative technology that is cheaper and more sustainable than traditional light sources, since it allows economic savings for poultry farming without significant alterations on the production parameters or the quality of the product.

## 1. Introduction

In addition to animal diet and genetics, the farming system is one of the factors that most influences the conversion efficiency of feed and therefore the production performance of poultry. The light (intensity, photoperiod, and wavelength) has direct effects on the behavior, physiology, immunity, and performance of poultry [1,2]. Artificial lighting is a fundamental tool used in poultry production that aims to improve food and water intake and consequently the growth and the economic yield of poultry [3]. The Council of the European Union has established that in poultry farms, the lighting must follow a 24 h rhythm and include periods of darkness lasting at least 6 h in total, with at least one uninterrupted period of darkness of at least 4 h, excluding dimming periods [4]. The impact of various intensities and wavelengths on the production performance of chickens has been extensively studied in recent decades; this has led to testing in commercial farms of various lighting systems, such as incandescent and fluorescent lamps. Recently, light-emitting diode (LED) lamps have gained increasing interest in poultry businesses due to their high energy efficiency, long operating life, availability in different wavelengths, low electricity consumption with consequent reduction of CO_2_ emissions into the atmosphere, and low farming costs [5]. Many studies have been conducted to evaluate the effect of monochromatic light produced by LED lamps on the production performance and quality of chicken meat, and although a wide range of colors is available, there are conflicting reports on the impact of different colors on production performance. It has been shown that LEDs, compared to other light sources, are able to reduce stress and fear in broilers [6]. Broilers reared under green or blue light showed significantly more body weight gain and muscle development compared to those reared under white and red light [1,7,8]. Cao et al. [7] correlated the increase of body growth with the stimulation of testosterone secretion and myofiber growth. In addition, an increase in body weight and carcass yield has been found when breeding poultry in the presence of LEDs, possibly due to muscle hypertrophy induced by the production of testosterone. According to a study of Parvin et al. [5], LEDs that emit blue, green, and yellow wavelengths are able to improve the immune system and meat quality in broilers. Blue and green light helps promote higher antibody production than red light. Poultry reared under mixed yellow and green-blue lights showed softer breast musculature, while white light resulted in increased amino acid content [5]. In a study carried out on chickens subjected to two-color green-blue LED lights for 81 d, chicks exposed to LED (green-blue) light gained weight compared to chicks exposed to normal artificial light without resulting changes in blood biochemical parameters. Moreover, in chickens under mixed green-blue light, the carcass yield and food conversion index were significantly higher than in chickens under single LED light [2]. Different light spectra produced by LED lights of 2700 K (Warm) or 5000 K (Cold) can influence the production, stress, and behavior of broilers; cold LEDs can reduce stress and fear, increasing weight and food conversion index [9].

At present, nutrient composition and meat quality characteristics of poultry products are widespread concerns by consumers. The quality of poultry products varies with growth rate and body composition [10]. Light has an influence on growth, since it can alter the structure of muscle myofiber, enhancing myoblast proliferation [11,12]. There is illimited information regarding LEDs and their impact on meat quality. However, there is no information in the literature on whether monochromatic light stimuli that showed growth-promoting effects can also influence breast meat composition and subsequent meat quality. Thus, the aim of this study was to investigate the effect of LED lights with three different color temperatures, Neutral (K = 3300–3700), Cool (K = 5500–6000), and Warm (K = 3000–2500), on production performance and breast muscle meat quality of broilers. It was concluded that the three light sources evaluated in this study may be suitable to replace the traditional light source in poultry facilities to reduce energy costs and optimize production efficiency.

## 2. Materials and Methods

### 2.1. Experimental Design and Samples Collection

The experimental test was carried out on a farm located in the Abruzzo region (Italy) that had adopted LED technology in breeding practices. All the procedures concerning the animals’ management were carried out in accordance with the European directive 2007/43/EC for the protection of chickens kept for meat production [4]. During the trial, no breeding practices other than those normally adopted were introduced; therefore, approval by the ethics committee was not considered necessary. The company made available 4 boxes, in one of which the conventional lighting system based on the use of neon was maintained (Control), and in the other three of which LED lights were installed in Neutral (Neutral LED; K = 3500–3700), Cool (Cool LED; K = 5500–6000), and Warm (Warm LED; K = 3000–2500) shades, respectively. The broilers (Ross 508, Aviagen Group, Huntsville, AL, USA) were exposed to light according to the European directive (6 h of darkness and 18 h of light). Male chicks, offspring of hens around 35 weeks of age, were vaccinated by local application for Marek’s disease, infectious bronchitis, Newcastle disease, and Gumboro disease. A total of 180 chicks were randomly divided into 4 groups (45 chicks per group) in 4 environmentally controlled rooms. Each room was divided in 3 replicates with 15 birds in each replicate. The production cycle lasted until the animals reached the commercial weight (3.30 ± 0.20 kg live weight). During the entire period, the animals were reared in accordance with the protocols normally applied by the company for the production of heavy chickens. All birds were fed the same diet throughout the study. Birds were provided a 3-phase feeding program (starter: 1 to 12 d; grower: 13 to 21 d; finisher: 22 to 48 d of age), the chemical characterization of which is reported in Table 1. Feed conversion ratio (FCR) could not be calculated because the boxes were equipped with a single silo from which the food was evenly distributed. At the end of the normal production cycle, 30 chickens from each experimental group were randomly selected, weighed, and slaughtered in the presence of the responsible veterinarian and in accordance with current regulations in terms of animal welfare. After evisceration, the carcasses were weighed in order to determine the yield. After 24 h, during which the carcasses were stored at 4 °C and covered by a synthetic film in order to avoid exposure to the surrounding environment, the hemibreast meat was sampled. Part of the meat was immediately used for the determination of pH, moisture, color, and drip loss, while the remainder was frozen at −20 °C for subsequent analysis (total lipids and fatty acid profile). The same samples used for drip loss, after 24 h, were used for cooking loss. The cooked meat was sampled at defined intervals to evaluate the volatile profile immediately after cooking (T0) and after 7 d (T7) stored at 4 °C.

### 2.2. Evaluation of Breast Meat Color

The pH evaluation on chicken breast samples 24 h (pH_24_) after slaughtering was performed by using a portable pH meter equipped with an electrode (Crison, Barcelona, Spain) that was inserted about 1–1.5 cm into the tissue, adjusting each evaluation in relation to the muscle temperature. Before the analysis, a calibration of the instrument was performed by using standard phosphate buffers (pH 4.00 and 7.00), and at the end of each measurement, the electrode was carefully rinsed in distilled water before the next evaluation. Color measurements were determined according to the procedure reported by Bennato et al. [13]. Briefly, the analysis was performed on the transverse section of the chicken breast muscle by using the Minolta CR-5 reflectance colorimeter. All evaluations were carried out taking into account the CIELAB system, which exploits the chromaticity coordinates L* (lightness), a* (redness), and b* (yellowness). Before each series of measurements, the colorimeter was calibrated by using a white tile (L* = 100) and black glass (a* = 0).

### 2.3. Drip Loss, Cooking Loss, and Chemical Composition of Breast Meat

In order to determinate the capability of meat to retain water, meat samples of about 2–2.5 cm of thickness and approximate weight of 100 g were inserted inside an expanded and closed bag and stored at 4 °C for 24 h. The meat was weighed at the beginning and end of this time period, and the drip loss was expressed as a percentage of the initial sample weight. The cooking loss, useful to characterize the ability of meat to retain water during cooking, was evaluated on the same meat samples used for drip loss. Meat samples were weighed and cooked in a water bath until the core temperature of 70 °C was reached. Samples were then cooled at room temperature and weighted. The cooking loss was expressed as a percentage of the initial raw sample weight. The evaluation of meat moisture and fat content was made according AOAC methods (2000) [14].

### 2.4. Fatty Acids Profile of Breast Meat

Fatty acids were extracted by using the Folch method (1957) [15]. Approximately 5 g of meat was homogenized by using Ultra-turrax-T25 with 45 mL of Folch solution (Chloroform: methanol, 2:1). Subsequently, the homogenized samples were transferred into a flat bottom flask and stirred in the dark for 7 h at room temperature. All samples were transferred into a separating funnel with the addition of 15 mL of 1% NaCl and left overnight. Total fat for each sample was obtained through the chloroform phase evaporation to dryness by using a Strike-Rotating Evaporator set at 40 °C. For each sample, the formation of fatty acid methyl esters (FAME) was induced by mixing 60 mg of fat with 1 mL of hexane and 500 µL of sodium methoxide. Detection of FAMEs was performed by a gas chromatograph (Focus GC; Thermo Scientific, Waltham, MA, USA) equipped with a capillary column (Restek Rt-2560 Column fused silica 100 m × 0.25 mm highly polar phase; Restek Corporation, Bellefonte, PA, USA) and a flame ionization detector (FID). Hydrogen was used as carrier gas. The initial holding temperature was 55 °C for 1 min; then it was increased to 170 °C at a rate of 10 C/min and held for 30 min. The final temperature of 215 °C was reached at a rate of 2 °C/min and held for 4 min. Peak areas were quantified using ChromeCard (Thermo Fisher Scientific, Milan, Italy) software, and the relative value of each individual FA was expressed as a percentage of the total FAME. The value of each FA was used to calculate the sum of saturated fatty acids (SFA), monounsaturated fatty acids (MUFA), and polyunsaturated fatty acid (PUFA).

### 2.5. Determination of Volatile Components of Cooked Breast Meat

Volatile compound (VOC) evaluation was performed with a gas chromatograph (Clarus 580; Perkin Elmer, Waltham, MA, USA) coupled with a mass spectrometer (SQ8S; Perkin Elmer, MA, USA) and equipped with an Elite-5MS column (length × internal diameter: 30 × 0.25 mm; film thickness: 0.25 µm; Perkin Elmer). Briefly, 3.5 g of minced cooked meat was mixed with 10 mL of an aqueous solution of NaCl (360 g/L) and 10 µL of internal standard (3-methyl-2-heptanone; 10 mg/kg in ethanol) and exposed to SPME fiber (divinylbenzene-carboxy-polydimethylsiloxane in solid phase; length: 1 cm, film thickness: 50/30 µm; Sigma-Aldrich, Milan, Italy) for 60 min at 60 °C. Then, the extracted VOCs were thermally desorbed in GC–MS. The thermal program and the recognition of the individual VOCs were performed as previously described by Ianni et al. [16].

### 2.6. Statistical Analysis

All the analyses were performed on 15 animals per group (5 samples for each replicate), randomly selected, and the analysis on the single sample was performed in triplicate. Results were reported as mean values with the correspondent standard deviations (SD). The statistical analysis was performed by using SigmaPlot 12.0 Software (Systat software Inc., San Jose, CA, USA) for windows operating system. The ANOVA model was applied, and the post-hoc comparison was performed through Tukey’s test; *p* values lower than 0.05 were considered statistically significant.

## 3. Results

### 3.1. Production and Physical Parameters

In Table 2, the productive parameters and the chemical characteristic of breast meat samples are reported. The live weight evaluations carried out at the end of the production cycle showed overlapping values among the Control, Neutral, and Cool LEDs. Significant variations were observed between Control and Warm LED (*p* < 0.01) and Cool and Warm LED (*p* < 0.05). A lower carcass weight was detected in the Warm LED group than in the Control group (*p* < 0.05). However, no significant variations among the groups were found in evaluations of carcass yield. Compared to the Control, the exposure to Neutral (*p* < 0.05) and Cool LEDs (*p* < 0.05) increased the ability of fresh meat samples to retain water. On the contrary, no significant differences were detected in meat samples exposed to Warm LED compared to the other treatment. The cooking loss values closed to 15–17% in all samples. Color evaluations showed a lower brightness in samples of meat belonging to chickens exposed to Warm LED compared to the other groups. Samples obtained from chickens reared under Neutral LED showed higher a* values compared to those from the Control. Within the groups exposed to different LEDs, significant chromatic variations in redness were observed between Neutral and Warm LEDs. A higher b* value was found in Neutral LED samples than in those from the other groups. Significant variations were observed in moisture between Neutral LED and Control. The lipid content of breast meat was found to be unaffected in all LED groups.

### 3.2. Fatty Acid Profile

The total lipid percentage is reflected in scarce variation in fatty acids profile (Table 3). The use of LEDs did not induce significant changes in the total content of SFA and monounsaturated fatty acids (MUFA). On the contrary, the exposure to Warm LED induced an increase of PUFA (*p* < 0.05) respect to the Control. In the meat samples belonging to the Neutral LED group, a significant increase compared to the Control was observed in cis-vaccenic acid (C18:1, cis11).

### 3.3. Volatile Profile of Cooked Meat

The evaluation of volatile profile in cooked chicken meat led to the identification of 18 compounds (both at T0 and T7) belonging to the family of aldehydes, alcohols, ketones, and phenolic compounds (Table 4). In all groups, the most represented compound was hexanal, the concentration of which did not significantly change between the groups at either T0 or to T7. Among alcohols, a significant decrease in 1-Octen-3-ol was recorded in T0 samples obtained from chickens reared under Neutral LED compared with the Control (*p* < 0.01). After 7 d of storage, compared to the Control, a significant increase of 1-Pentanol (*p* < 0.05) was detected in Neutral LED samples; conversely, in Cool LED samples, 1-Octanol (*p* < 0.05) and 1-Octen-3-ol (*p* < 0.05) decreased. Regarding ketones, significant decreases related to LED light exposure were observed for all identified compounds. In particular, significant variations were observed in 2-Heptanal (*p* < 0.05) between the Cool and Warm groups, in 1-Octenal (*p* < 0.05) between the Neutral and Warm groups, in 2-Hexanone, 4-methyl (*p* < 0.05) between the Control and Neutral groups, and in 3-Octanone, 2-methyl (*p* < 0.05) between the Control and Warm groups. On the contrary, after 7 d, 1-Octenal differed between Cool and the other groups, and 3-Octanone, 2-methyl differed between Control and Cool and Control and Warm. No significant changes within the groups were observed for the phenolic compounds.

## 4. Discussion

One of the biggest challenges in broiler production is to achieve maximum production while reducing energy consumption and production costs. Artificial lighting is a fundamental tool used in poultry production that aims to improve food and water intake and consequently the growth and the economic yield of poultry. It has been shown that the use of LED light in poultry farms has numerous advantages linked to greater energy savings, greater luminous efficiency, and a reduction in environmental impact that can nevertheless improve the production parameters of broilers.

Our study showed that Neutral and Cool LED light did not determine changes in broiler growth; on the contrary, broilers reared under Warm LED had a lower live weight than those reared under the Control and Cool LED, though there was no influence on the carcass yield. Previous studies carried out on broiler growth performance and development have showed discordant results [9,17,18]. Archer (2017) [9] observed that broilers raised under Cool LED (K = 5000 K) grew to a heavier weight at the end of 42 d than birds raised under Warm LED (K = 2700). Moreover, Cool birds had better FCR than Warm birds. These results demonstrate that raising broilers under 5000 K LED lights can reduce their stress and fear and increase weight gain when compared with 2700 K lights. In Arbor Acres broilers, blue monochromatic light increased body weight and carcass yield compared with red, white, and green light [19]. On the contrary, ducks reared under red light showed an increase of body weight, body weight gain, and feed intake compared to those reared under yellow light [20]. In broilers exposed for two weeks to LED with different wavelengths, it has been demonstrated that green light increased the mRNA and protein levels of growth hormone-releasing hormone (GHRH) in the hypothalamus as well as plasma growth hormone (GH) concentrations by activating the secretion of plasma melatonin, which plays a key role in photoelectric conversion [21]. These findings suggest the capability of light to modulate gene expression. The differences observed in our experimental condition may be due to the use of differing LED light sources, varying light intensities, time of exposure, and animal species.

Beyond broiler performance and meat yield, an important aspect, especially for consumers, is meat quality. Therefore, in our study, we evaluated the possible effects of LED light on quality properties of breast meat such as water-holding capacity, cooking loss, lightness, fat content, and the fatty acids profile and volatile profile of cooked meat.

The exposure to LED affected the ability of raw meat to retain water (drip loss); Neutral and Cool LED meat samples showed a greater weight loss compared to both the Control and Warm LED samples. In a study conducted by Wang et al. [22], an association was demonstrated in goat *longissimus dorsi* muscle between high drip loss values and low levels of metabolic enzymes (α-enolase, NADH dehydrogenase, pyruvate dehydrogenase), stress response factors (Hsp27), and structural proteins (myosin) that affected glycolysis, oxidation, and muscle contraction. The authors showed that the correlation between the declines in α-enolase, NADH dehydrogenase, and pyruvate dehydrogenase levels and higher drip loss may be due to an increase of glycolysis and accumulation of lactic acid in muscle tissues with consequent lowering of pH. The pH decline to the pI value of myosin reduces the distance between thick and thin myofilaments and the sarcomere length [23], allowing water in the myofibril gap to flow out with the consequence of an increase of drip loss values. However, in our experimental condition, the increase of drip loss values observed in Neutral and Cool LED was not associated to a decrease in pH, which may owe to different characteristics of the myofiber of breast muscle. The relationship among muscle fiber characteristics, size, types with post-mortal biochemical processes, and meat quality is well documented in different animal species [24,25]. In the *longissimus dorsi* muscle of bulls, a correlation has been observed between muscle fiber area and water-holding capacity indicating that muscles with larger fiber areas had a lower drip and ageing loss but a higher cooking and grilling loss [26]. In a study by Cao et al. [7] carried out on broilers reared under different light spectra, it was observed that the cross-section area and density of myofibers changed among various groups, and the myofiber area of blue-light-exposed broilers was larger than that of the other group, suggesting that light could influence the myofiber growth in broiler skeleton muscle and consequently water holding capacity. No significant change among the groups was observed in cooking loss, and it may be correlated to the same fat content in breast muscle samples of the different treatment. In duck breast meat, higher cooking loss values have been observed in breast muscle containing high lipid levels [27].

Meat color greatly influences choices made by the consumer. The factors that can influence the color are the content of pigments (myoglobin and hemoglobin), the time elapsed since slaughter, the processing and storage conditions, the muscle section considered, the cutting direction, the fat, the state of hydration, and the species, sex, and age of the animal. In this study, significant differences were evidenced among the different treatments for lightness (L*) and for chromaticity coordinates a* (redness) and b* (yellowness). The normal color of raw chicken breast is slightly pinkish but can also appear bluish-white to yellow; it depends on the concentration and the chemical and physical state of myoglobin and also on the structure of the meat surface. In all the groups, meat samples had low values of lightness with respect to values reported in literature; however, as reported by Petracci et al. [28], color variation of broiler breast meat could be affected by the age and season of broiler slaughter. Our data showed a lower L* value in Warm LED breast meat samples than in other groups. Our data are in contrast with other studies that showed higher L* values in the breast of broilers exposed to red light than in white, green, and blue light [19]. Napper et al. [29] found heat stress to cause a significant increase in the lightness of broiler chicken meat compared with a cold stress treatment. Schneider et al. [30] reported that meat from the hot and thermo-neutral treatments was lighter in color than that from the cold group. An increase in the L* value is associated to the denaturation of myofibrillar proteins, followed by aggregation, consequently changing the surface reflectance and increasing lightness [31]. Therefore, a lower lightness in the Warm breast sample may be associated to a different composition in myofibrillar proteins. Regarding the a* and b* parameters, differences were observed among all the groups. Meat redness is associated to the oxidation state of hemoglobin; a decrease of redness is attributed to a large degree to the oxidation of the bright red oxymyoglobin or the purplish deoxymyoglobin into the brownish metmyoglobin, as well as to the denaturation of myoglobin. In research by Lindahl et al. [32] on pork meat, it was evidenced that about 86–90% of variations in meat lightness, redness, yellowness, and chroma (saturation) can be explained by variations of pigment content, myoglobin forms, and reflectance of internal surface. Specifically, lightness, redness, and chroma appear to be influenced to a similar extent by both the pigment content and the myoglobin forms, while yellowness seems to vary mostly in relation to the myoglobin forms and less to internal reflectance, without significant effects induced by pigment content. For this reason, the observed finding deserves further and more specific assessments. Broilers reared under Neutral LED had a significant increase of cis-vaccenic acid (C18:1, cis 11). Cis-vaccenic acid can derive from the diet or biosynthetic pathways, but its role in metabolism is largely unknown. In mammalians, it has been confirmed that 18:1, cis-11 is an elongation product of 16:1 by the ELOVL6 isoform of the mammalian elongase enzyme, which adds an acetate molecule to the carboxylic acid end of the fatty acyl chain [33]. An increase of total PUFA was observed in meat of broilers reared under Warm light; however, the ratio of PUFA/SFA did not change in any of the different experimental conditions. There are few reports of the effects of light color on muscular fatty acid composition. Kim et al. [34] reported that red light LED increased the concentration of MUFA, SFA, and the SFA/PUFA ratio, but reduced the concentration of PUFA, n-3 fatty acid, and n-6 fatty acid, and it is still unclear exactly how light color alters meat fatty acid composition. It is possible to hypothesize an influence of light on gene expression; in fact, exposure to light can alter metabolic function, and many genes involved in nutrient metabolism display rhythmic oscillations. Therefore, further study is required to evaluate the influence of light on fatty acid metabolism in broilers.

The characterization of volatile compounds in foods represents an aspect of pivotal interest for understanding the biochemical mechanisms responsible for the accumulation of flavor determinants. These mechanisms are mostly associated with the degradation of the lipid and protein components and are very often mediated by endogenous enzymatic forms or by oxidative events induced by the heating treatments used for cooking. With specific regard to cooked meat, the accumulation of volatile flavor compounds is generally due to reactions that concern above all thermal lipid degradation and Maillard–lipid interactions [35]. Obviously, the accumulation of certain VOC depends on the type of substrate present in raw meat, which in turn is a function of several pre- and post-slaughter factors, including breed, feeding strategy, post-mortem carcass conservation, and cooking method.

In this study, 18 VOC were detected in cooked meat samples obtained from all the experimental groups. The majority of these compounds were aldehydes, alcohols, and ketones, i.e., chemical families mainly deriving from the lipolytic process. Such findings are therefore inserted in a context of compliance with respect to what has been reported in other studies, in which the main substrate for the production of flavor compounds in cooked poultry meat is highlighted in the lipid component [36]. Immediately after cooking and after 7 d of storage of the cooked product, the most represented compound was hexanal. This finding was quite expected, since hexanal is a volatile compound characteristic of cooked poultry meat [37]. Furthermore, hexanal is also generally considered a marker of lipid oxidation, and the lack of variations between the different analyzed conditions testifies to the fact that the lighting program did not induce effects from this point of view. Specifically, this may be due to the presence of similar contents of linoleic acid (C18:2), which is the main substrate for hexanal production. A compound to which significant differences were associated among the various experimental groups was 1-octen-3-ol, belonging to the alcohols group. This compound provided fishy, fatty, mushroom, grassy odors, and was mainly derived from enzymatic reactions mainly catalyzed by lipoxygenases and hydroperoxide lyases. High values of this compound have been associated with marked oxidative processes, so it is telling that immediately after cooking, the samples obtained following the breeding with Neutral and Warm LED were found to be poorer in this compound. In a study conducted by Mielnik et al. [38] on cooked turkey meat, a direct proportionality was highlighted between the 1-octen-3-ol concentration and the presence of thiobarbituric acid-reactive substances (TBARS). Therefore, cooked meat low in this compound was associated with better resistance to lipolytic processes, with a consequent improved chemical stability of the product over the storage period. With regard to meat samples analyzed after storage, the data was confirmed only for samples obtained from the treatment with Warm LED, which therefore tends to confirm itself as the best condition for limiting the oxidative processes.

Exposure to Neutral and Warm LEDs also proved effective in limiting the accumulation of ketones immediately after cooking, while the treatment with Cool LED produced values comparable to those obtained from the Control group. These compounds are reported to directly derive from hydroperoxides, which are considered to represent the first oxidation products [39,40]. Hydroperoxides are described to be odorless, with a slight possibility of contributing to meat aroma; however, the ketone accumulation was instead associated with the onset of off-flavors and off-odors. The data obtained in the presence of treatment with Neutral and Warm LED were therefore confirmed of potential interest, although it should be considered that after 7 d of cooked meat storage this evidence disappeared, obtaining values comparable to those of the Control group.

## 5. Conclusions

The application of three shades of LED lights in broilers did not result in significant changes in carcass yield. Even the parameters evaluated on the breast muscle did not reveal any changes that could justify a qualitative alteration of the product. The most interesting data concerns the volatile profile characterized on cooked meat samples. This approach indicated an improvement, albeit very slight, in the oxidative stability of the samples obtained from the Neutral LED group. These changes in the volatile profile are also generally associated with variations in the aroma and taste of food products, so it would be desirable to develop sensory assessments, based above all on the organization of panel tests, in order to better characterize this aspect. Overall, therefore, the application of LED light in broiler breeding has led to production that is qualitatively in line with the company high standards but more sustainable both from an economic and environmental point of view, since LED is a technology with less impact on energy consumption and therefore on the release of CO_2_.

## Figures and Tables

**Table 1 animals-11-01505-t001:** Chemical composition of basal diet for broilers.

Chemical Composition (%)	Starter	Grower	Finisher
Dry Matter	89.10	88.90	89.00
Crude Protein ^†^	25.45	21.70	19.65
Ether Extract ^†^	5.65	6.88	7.97
Crude Fiber ^†^	3.54	3.64	3.82
Ash ^†^	6.23	5.71	4.78
Lysine ^†^	1.54	1.23	1.17
Methionine ^†^	0.63	0.46	0.39
Sodium ^†^	0.17	0.16	0.15
Calcium ^†^	0.85	0.74	0.65
Phosphorus ^†^	0.68	0.56	0.45

^†^ All data are reported on a dry matter basis.

**Table 2 animals-11-01505-t002:** Physical and chemical characterization of breast meat samples obtained from broilers exposed to different light treatments.

Trait	Control	Neutral LED	Cool LED	Warm LED
Live weight, kg	3.53 ± 0.29 ^a^	3.48 ± 0.27 ^abc^	3.52 ± 0.29 ^ab^	3.22 ± 0.39 ^c^
Carcass weight, kg	2.58 ± 0.25 ^a^	2.56 ± 0.24 ^ab^	2.61 ± 0.25 ^ab^	2.38 ± 0.31 ^b^
Carcass yield, %	73.85 ± 11.43	75.17 ± 9.42	74.81 ± 10.47	75.46 ± 16.84
Drip loss, %	0.90 ± 0.14 ^a^	1.22 ± 0.43 ^b^	1.10 ± 0.29 ^c^	1.08 ± 0.34 ^abc^
Cooking loss, %	16.37 ± 2.30	17.73 ± 3.35	16.22 ± 4.05	14.96 ± 2.20
pH_24-h_	5.69 ± 0.07	5.64 ± 0.07	5.73 ± 0.07	5.68 ± 0.06
**Color**				
L*	27.92 ± 3.03 ^a^	28.86 ± 2.95 ^a^	28.19 ± 3.56 ^a^	26.44 ± 2.17 ^b^
a*	−1.57 ± 0.46 ^a^	−1.90 ± 0.34 ^b^	−1.69 ± 0.53 ^ab^	−1.43 ± 0.38 ^a^
b*	0.26 ± 0.99 ^a^	1.27± 1.59 ^b^	0.23 ± 1.14 ^ac^	0.77 ± 0.62 ^ad^
**Chemical Composition (%)**				
Moisture	73.21 ± 2.77 ^a^	75.86 ± 1.35 ^b^	74.58 ± 0.93 ^ab^	74.38 ± 2.00 ^ab^
Dry matter	26.79 ± 2.77 ^a^	24.15 ± 1.35 ^b^	25.42 ± 0.93 ^ab^	25.62 ± 2.00 ^ab^
Total lipids ^†^	6.34 ± 2.19	6.25 ± 1.69	6.08 ± 1.73	5.72 ± 1.37

Data are reported as mean ± standard deviation (SD); ^a–d^ Different letters in the same row indicate significant differences (*p* < 0.05). pH_24-h_: pH 24 h post-mortem. L*: lightness; a*: redness; b*: yellowness. ^†^ Data are reported on a dry matter basis.

**Table 3 animals-11-01505-t003:** Fatty acid profiles of breast meat samples obtained from broilers exposed to different light treatments.

Fatty Acids	Control	Neutral LED	Cool LED	Warm LED
C14:0	0.47 ± 0.11	0.53 ± 0.13	0.43 ± 0.06	0.45 ± 0.09
C15:0	0.05 ± 0.03	0.07 ± 0.03	0.05 ± 0.02 ^b^	0.05 ± 0.01
C16:0	19.52 ± 1.87	20.13 ± 1.48	20.01 ± 2.17	19.83 ± 1.49
C17:0	0.17 ± 0.05	0.19 ± 0.04	0.16 ± 0.02	0.17 ± 0.02
C18:0	8.72 ± 1.93	9.27 ± 1.61	8.89 ± 1.23	8.04 ± 1.90
C20:0	0.08 ± 0.01	0.07 ± 0.01	0.06 ± 0.03	0.10 ± 0.10
C14:1	0.02 ± 0.01	0.03 ± 0.02	0.03 ± 0.01	0.03 ± 0.02
C16:1	1.83 ± 0.58	1.72 ± 0.87	1.86 ± 0.39	2.14 ± 0.79
C18:1, c9	23.56 ± 2.69	23.82 ± 3.65	23.72 ± 2.24	23.99 ± 2.64
C18:1, c11	1.35 ± 0.29 ^a^	1.59 ± 0.21 ^b^	1.55 ± 0.40 ^ab^	1.47 ± 0.18 ^ab^
C22:1	0.13 ± 0.07	0.12 ± 0.04	0.12 ± 0.05	0.12 ± 0.06
C18:2	34.56 ± 3.40	32.72 ± 5.33	33.12 ± 3.18	34.66 ± 1.74
C18:3	3.35 ± 0.72	3.19 ± 0.87	3.26 ± 0.60	3.37 ± 0.42
C20:4	3.67 ± 1.97	3.98 ± 1.88	4.07 ± 1.45	3.20 ± 1.88
Others	2.52 ± 0.61	2.58 ± 0.56	2.66 ± 0.54	2.39 ± 0.59
SFA	29.01 ± 3.52	30.26 ± 2.27	29.61 ± 3.03	28.64 ± 2.38
MUFA	26.90 ± 2.86	27.28 ± 4.61	27.28 ± 2.44	27.75 ± 3.25
PUFA	39.04 ± 3.84 ^a^	39.89 ± 5.79 ^ab^	40.45 ± 2.81 ^ab^	41.23 ± 2.11 ^b^
UFA/SFA	2.32 ± 0.43	2.25 ± 0.24	2.32 ± 0.33	2.43 ± 0.31

Data are reported as mean ± standard deviation (SD); ^ab^ Different letters in the same row indicate significant differences (*p* < 0.05). SFA: saturated fatty acids; MUFA: monounsaturated fatty acids; PUFA: polyunsaturated fatty acids.

**Table 4 animals-11-01505-t004:** Volatile profiles of cooked breast meat samples obtained from broilers exposed to different light treatments.

VOC	T0				T7			
	Control	Neutral LED	Cool LED	Warm LED	Control	Neutral LED	Cool LED	Warm LED
**Aldehydes**								
Pentanal	2.35 ± 0.22	4.14 ± 2.16	2.59 ± 1.76	1.56 ± 1.10	1.06 ± 0.17	2.15 ± 0.26	1.55 ± 0.24	1.74 ± 0.12
Hexanal	74.48 ± 0.05	71.24 ± 1.13	72.03 ± 4.82	79.91 ± 6.45	62.68 ± 1.55	52.23 ± 4.98	62.29 ± 3.34	67.21 ± 4.22
Heptanal	2.02 ± 0.23	6.87 ± 6.52	1.43 ± 2.02	4.33 ± 4.98	3.09 ± 0.22	3.17 ± 0.24	1.99 ± 0.30	1.98 ± 0.28
Octanal	1.81 ± 0.07	2.64 ± 1.01	1.99 ± 0.62	1.96 ± 0.67	2.07 ± 0.15	2.60 ± 0.33	2.04 ± 0.12	2.10 ± 0.17
Nonanal	1.86 ± 0.11	2.55 ± 0.54	2.85 ± 0.20	2.18 ± 0.49	2.78 ± 0.40	3.01 ± 0.31	2.79 ± 0.31	2.89 ± 0.30
**Alcohol**								
1-Pentanol	1.58 ± 0.17	1.43 ± 0.55	2.35 ± 0.49	1.56 ± 0.03	2.00 ± 0.24 ^a^	1.93 ± 0.26 ^ab^	2.54 ± 0.32 ^ab^	2.83 ± 0.26 ^b^
1-Heptanol	0.17 ± 0.08	0.23 ± 0.13	0.21 ± 0.30	0.14 ± 0.19	0.90 ± 0.15	0.98 ± 0.12	0.70 ± 0.13	0.68 ± 0.13
1-Octanol	nd	0.07 ± 0.10	nd	nd	0.45 ± 0.06 ^a^	0.52 ± 0.08 ^ab^	0.28 ± 0.09 ^b^	0.20 ± 0.06 ^ab^
1-Octen-3-ol	8.20 ± 0.05 ^a^	5.53 ± 0.11 ^b^	9.36 ± 1.34 ^ab^	2.83 ± 2.06 ^ab^	10.61 ± 0.95 ^a^	21.44 ± 1.94 ^ab^	9.62 ^b^ ± 1.02	8.70 ± 0.84 ^ab^
2-Octen-1-ol	0.39 ± 0.04	0.20 ± 0.10	0.25 ± 0.36	0.03 ± 0.04	1.34 ± 0.40	0.86 ± 0.14	1.05 ± 0.14	0.89 ± 0.11
1-Hexanol, 2-ethyl-	0.14 ± 0.20	0.75 ± 0.24	1.54 ± 1.37	1.73 ± 1.01	nd	0.07 ± 0.10	nd	nd
**Ketones**								
2-Heptenal	0.90 ± 0.37 ^ab^	0.28 ± 0.03 ^ab^	0.65 ± 0.04 ^a^	0.16 ± 0.07 ^b^	0.74 ± 0.09	0.34 ± 0.05	0.68 ± 0.07	0.57 ± 0.08
1-Octenal	1.52 ± 0.32 ^ab^	0.88 ± 0.02 ^a^	1.50 ± 0.57 ^ab^	0.56 ± 0.05 ^b^	0.51 ± 0.08 ^a^	0.62 ± 0.10 ^a^	1.37 ± 0.03 ^b^	0.84 ± 0.17 ^a^
2-Hexanone, 4-methyl-	0.08 ± 0.01 ^a^	0.01 ± 0.01 ^b^	0.04 ± 0.06 ^ab^	0.02 ± 0.03 ^ab^	0.24 ± 0.02	0.19 ± 0.12	0.25 ± 0.17	0.23 ± 0.19
3-Octanone, 2-methyl-	3.27 ± 0.29 ^a^	2.15 ± 0.83 ^ab^	2.20 ± 0.32 ^ab^	1.68 ± 0.13 ^b^	3.95 ± 0.21 ^a^	4.03 ± 0.37 ^abc^	3.18 ± 0.37 ^b^	3.04± 0.27 ^c^
**Phenolic Compounds**								
Ethylbenzene	0.50 ± 0.17	0.41 ± 0.10	0.48 ± 0.14	0.71 ± 0.15	3.13 ± 0.46	2.40 ± 0.40	4.27 ± 0.38	2.71 ± 0.31
*p*-Xylene	0.63 ± 0.13	0.53 ± 0.05	0.43 ± 0.34	0.54 ± 0.19	4.25 ± 0.17	3.27 ± 0.28	5.05 ± 0.57	3.03 ± 0.69
Benzaldehyde	0.10 ± 0.08	0.13 ± 0.07	0.09 ± 0.07	0.13 ± 0.01	0.20 ± 0.17	0.25 ± 0.04	0.34 ± 0.04	0.35 ± 0.05

Data are reported as mean percentage of each volatile compound (VOC) ± standard deviation (SD). ^abc^ Different letters in the same row indicate significant differences (*p* < 0.05).

## Data Availability

The data presented in this study are available on request from the corresponding author.

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
