# Peer review of "Evaluation of Chemical Composition and Meat Quality of Breast Muscle in Broilers Reared under Light-Emitting Diode"

_animals, 2021, doi:10.3390/ani11061505_

Round 1
Reviewer 1 Report
The aim of this study was to determine the influence of LED light on growth performance and meat quality traits on broilers. The number of broiler meat samples used in the experiment is sufficient, the study methods used are correct. The result is well carried out. Before publishing in animals, the paper requires additions and corrections. The list of proposed changes is given below:
- Line 32-33 author mentioned ‘LED represents an alternative technology cheaper and more sustainable than the traditional light.’ The Economic and environment benefits on poultry farm can be added in discussion area.
- Line 123, why authors evaluate the volatile profile after 7 ds (T7)?
- Line 129, pH24h
- Line 191, 198, 199.., P value
- Line 195, Sentence is grammatically incorrect.
- Line 214, Table 2, chemical composition, why 73.21 mark ‘a’, while 75.86 mark ‘b’
- Line 260, Table 4, 3-Octanone, 2-methyl, the abcd is confusing
- Line 267, however
- Authors could add more discussions the of influence of LED light on behavior and physiology in this study.
Author Response
The aim of this study was to determine the influence of LED light on growth performance and meat quality traits on broilers. The number of broiler meat samples used in the experiment is sufficient, the study methods used are correct. The result is well carried out. Before publishing in animals, the paper requires additions and corrections. The list of proposed changes is given below:
- Line 32-33 author mentioned ‘LED represents an alternative technology cheaper and more sustainable than the traditional light.’ The Economic and environment benefits on poultry farm can be added in discussion area.
- Thank you for the suggestion, this aspect has been added in the Discussion (Lines 270-276).
- Line 123, why authors evaluate the volatile profile after 7 ds (T7)?
- The evaluation of the volatile profile was performed after 7 ds in order to obtain more information related to the development of volatile compound as consequence of the oxidative process.
- Line 129, pH24h
- The manuscript has been corrected (Line 129).
- Line 191, 198, 199.., P value
- It has been corrected in the whole manuscript.
- Line 195, Sentence is grammatically incorrect.
- The sentence has been corrected (Lines 198-199).
- Line 214, Table 2, chemical composition, why 73.21 mark ‘a’, while 75.86 mark ‘b’
- The data have different letter because there is a significant difference between Control and Neutral LED.
- Line 260, Table 4, 3-Octanone, 2-methyl, the abcd is confusing.
- The Table 4 has been corrected.
- Line 267, however
- The Manuscript has been corrected (Line 278).
- Authors could add more discussions the of influence of LED light on behavior and physiology in this study.
- These aspects were not investigated because they did not fall within the aim of the study. However, it could be interesting to investigate the effects of the LED on behavior and better understand the molecular mechanisms underlying the observed changes in breast muscle.

Reviewer 2 Report
It is necessary to improve the statistical part. How many repetitions for each type of light and how many repetitions for each experimental unit considered. The quality of the chicken breast is influenced by nutrition and little importance was given to the composition of the feed. Total amino acids are not used, but digestible. You also need to know the total sulfur amino acids (Met+Cys).

Author Response
It is necessary to improve the statistical part. How many repetitions for each type of light and how many repetitions for each experimental unit considered. The quality of the chicken breast is influenced by nutrition and little importance was given to the composition of the feed. Total amino acids are not used, but digestible. You also need to know the total sulfur amino acids (Met+Cys).
Dear Reviewer,
thank you for your suggestions. The statistical part has been improved in the manuscript (Lines 189-192). In the present study all the animals received a standard diet, whose composition was defined taking into account the nutritional requirements indicated by the National Research Council. Since the aim of the work was to evaluate the effects of LED light on the quality of the meat, in order not to add further variables, the diet used by the feeder was respected and no changes were made to the diet. The ingredients and quantities shown are in accordance with nutritional needs. No analysis was made on the amino acid composition because it was not part of our objectives so only the values ​​of methionine and cysteine ​​are reported. However, if the reviewer considers it necessary, we may add the requested information.
- Marek cannot be applied by spray. this information is not correct. One-day-old vaccination.
- The sentence has been corrected in the Manuscript (Line 104).
- How many birds per group. if it is 45 for 4 groups would give 11.25 each?? Improve writing. He's confused.
- This part has been improved (Lines 105-106; Lines 189-192).
- Live weight, kg 3.53 ± 0.29a 3.48 ± 0.27abc 3.52 ± 0.29ab 3.22 ± 0.39c is correct?
- The data are corrected and the Conclusion has been modified (Line 428)
- Hasn't CO2 production been measured for each light source??
- In our study we have not measured CO2

Reviewer 3 Report
Dear Editor,
The study investigated the effect of different LED colour temperatures on growth performance, carcass characteristics and breast meat quality of broilers, but the topic doesn't reflect these parameters. Although the study is interesting, the number of replications used is too low which raises questions on the credibility of the results. In addition, the study involved the rearing and slaughter of the birds, thus the authors should have sought for ethical clearance. The lack of ethical approval is a major concern even if breeding was not performed considering that the different lights could impact on the welfare of the birds. Also, modern farm practises involves slaughtering the birds from 35 - 42 days, it is not clear why the authors went as far as 48 days. The experimental design has a lot of techincal deficiencies. Some methods require references e.g. drip loss, cooking loss... Langauge editing can also improve the readablity of the paper. The conclusions are a bit far-fetched, it can be improved by pre-empting the results.
Thank you for the opportunity to review.
With regards
Author Response
Dear Editor,
The study investigated the effect of different LED colour temperatures on growth performance, carcass characteristics and breast meat quality of broilers, but the topic doesn't reflect these parameters. Although the study is interesting, the number of replications used is too low which raises questions on the credibility of the results. In addition, the study involved the rearing and slaughter of the birds, thus the authors should have sought for ethical clearance. The lack of ethical approval is a major concern even if breeding was not performed considering that the different lights could impact on the welfare of the birds. Also, modern farm practises involves slaughtering the birds from 35 - 42 days, it is not clear why the authors went as far as 48 days. The experimental design has a lot of technical deficiencies. Some methods require references e.g. drip loss, cooking loss... Langauge editing can also improve the readablity of the paper. The conclusions are a bit far-fetched, it can be improved by pre-empting the results.
Thank you for the opportunity to review.
With regards
Dear Reviewer,
thank you for the comments. Our experimental design does not differ much from that used in other works (i.e. Kim et al, 2013 [20]; Kim et al., 2014 [34]. In Italy, the slaughter of broilers can take place at different ages in relation to the destination: light chicken slaughtered at the age of 28-30 days (1.6 kg live weight); medium, slaughtered at 38-40 days (2.6 kg live weight) and heavy, males slaughtered at 48-50 days (3.5-3.6 kg live weight). The study was performed in a commercial poultry farm that followed the guidelines for breeding heavy broiler. All the procedures concerning the animals’ management were carried out in accordance with the Italian legislation (D.Lgs. 267/2003 of the Italian Parliament). During the trial, no breeding practices other than those normally adopted were introduced, therefore approval by the ethics committee was not considered necessary. At the end of the experimentation, all the samples were taken in a commercial abattoir where the chickens were slaughtered. No changes were made to the diet and farming system and no treatment was carried out on the chickens. The methods used for drip loss and cooking loss were the same used in our previous studies (i.e. Bennato et al., 2020 [13]).
